# Interventions Designed to Support Physical Activity and Disease Prevention for Working from Home: A Scoping Review

**DOI:** 10.3390/ijerph20010073

**Published:** 2022-12-21

**Authors:** Melanie Crane, Alec Cobbold, Matthew Beck, Tracy Nau, Christopher Standen, Chris Rissel, Ben J. Smith, Stephen Greaves, William Bellew, Adrian Bauman

**Affiliations:** 1Prevention Research Collaboration, Sydney School of Public Health, The Charles Perkins Centre, The University of Sydney, Camperdown, Sydney, NSW 2006, Australia; 2Institute of Transport and Logistics Studies, The University of Sydney Business School, The University of Sydney, Darlington, Sydney, NSW 2006, Australia; 3Centre for Primary Health Care and Equity, School of Population Health, UNSW Sydney, Sydney, NSW 2052, Australia; 4Health Equity Research and Development Unit, Sydney Local Health District, Camperdown, Sydney, NSW 2006, Australia; 5College of Medicine and Public Health, Flinders University, Sturt Rd, Adelaide, SA 5024, Australia; 6Western Sydney Local Health District, Westmead Hospital, Westmead, Sydney, NSW 2145, Australia

**Keywords:** working from home, teleworking, physical activity, workplace health promotion, sedentary behavior, chronic diseases, prevention, review, health policy, COVID-19, built environments

## Abstract

Working from home (WfH) has public health implications including changes to physical activity (PA) and sedentary behavior (SB). We reviewed published and grey literature for interventions designed to support PA or reduce SB in WfH contexts. From 1355 published and grey literature documents since 2010, we screened 136 eligible documents and extracted ten intervention studies. Interventions designed specifically for WfH were limited and included structured exercise programs, infrastructure (e.g., sit-stand workstations), online behavioral and educational programs, health professional advice and peer support, activity trackers and reminder prompts. Evidence of interventions to improve PA and reduce SB in WfH contexts is emergent but lacking in variety and in utilization of local environments to promote good health. Evidence is needed on the adaptation of existing workplace interventions for home environments and exploration of opportunities to support PA through alternative interventions, such as urban planning and recreational strategies.

## 1. Introduction

The COVID-19 pandemic has changed the way people work. *Work(ing)-from-home (WfH)* is now more common, particularly in office-based professions [1]. Pre-pandemic, WfH was limited to few organizations and individuals. In the US, 5% of the workforce were working from home in 2018, increasing to 45% during the early phase of the pandemic. Around 21% are expected to continue doing so beyond 2022 [2]. Similar patterns have been reported in Europe [3], while in Australia, 41% of the workforce WfH compared to 8% pre-pandemic [4]. Many organizations have embedded working from home within *flexible work arrangements* as an outcome of the conditions of the pandemic [4]. Now as employers, governments, and urban planners grapple with the implications of working from home on office space, workplace policies, digital technologies and business district planning [5], they need to also consider potential health implications.

Workplace settings are important for the prevention of non-communicable disease (NCD) and to promote healthy lifestyles [6]. Workplace health promotion programs may comprise policies, environmental supports, and activities to reduce lifestyle risk factors for NCDs; these differ from work/occupational health and safety programs which focus on minimizing risk of injury and illness inherent in workplace operations [6]. As employees adapt to WfH, existing health promotion programs may no longer suffice for NCD prevention. To ensure that preventive health strategies are suitable in the context of WfH changes, an examination of workplace health promotion strategies are needed.

Recent research suggests positive health benefits associated with WfH, including reduced work-family conflict, lower absenteeism, better psychological health, and better physical health [7,8,9]. There are mixed findings on the impact of WfH on physical activity (PA). Some studies which guided early pandemic guidance suggested PA was higher amongst those working from home [10], but more recent studies report decreases in PA and increases in sedentary behaviors (SB) [9,11]. Pre-COVID, some 25% of the world’s population were already failing to achieve sufficient PA as recommended by the World Health Organization (WHO) [12]. High-income countries, where desk-based workers are more prevalent, are most at risk of physical inactivity [1,12]. Physical inactivity is estimated to cost the global economy tens of billions of dollars in healthcare expenditure and lost productivity [13]. Unless it is better understood and addressed, the negative impact of WfH on PA could compound an already substantial health and economic burden.

Workplaces remain a key intervention setting for PA [14]. Interventions designed to improve PA have targeted workplace infrastructure (e.g., stair use, sit-stand desks) or structured programs, awareness raising, behavioral support (e.g., group-based exercise programs, counselling) and commuting (e.g., active travel) [15,16,17,18]. For workplace health promotion programs in the WfH context, targeting the journey to work or in-office programs, is arguably insufficient as a strategy for promoting PA. In addition, SB may increase without mechanisms to interrupt sitting [9]. Key interventions and policies for PA in the workplace need to be re-designed for WfH. The purpose of this review was to identify existing and emerging evidence-based interventions for PA in the WfH context.

## 2. Methods

### 2.1. Literature Review Search Strategy

A scoping review design was deemed most appropriate for extracting information from new and unpublished sources, given the recent changes in the WfH context. We applied the Joanna Briggs Institute framework for scoping reviews, developing an a priori protocol for framing the research question, identification and selection of studies, search, extraction and analysis of evidence as outlined below [19]. The primary research question was ‘which workplace health promotion interventions have been developed to support PA for the working from home environment.’ Searches for published literature were undertaken in Scopus, Web of Science and various Ebsco Host databases, while searches for grey literature were undertaken in Google, Mednar (a deep web search portal for medical research) and targeted websites. Search terms included a combination of WfH terms, PA/SB terms, and health intervention terms. From the Google search the first five pages of results were exported, following recommended practice [20]. Targeted websites included international health authorities and business consulting companies identified in consultation with experts. Table 1 provides a summary of the search strategy. See Appendix A for full details.

### 2.2. Document Selection

After removing duplicates, we screened titles and abstracts using the following criteria.

Inclusion criteria:Scientific studies, case studies, reports, book chapters, reviews that described an intervention aiming to improve PA or interrupt SB for office workers in a home-based setting. “Work” was defined as time spent in paid work from home for any number of days/hours over the working week.The intervention included any form of activity undertaken during work time, or to replace active commuting time that could contribute to a person achieving sufficient PA to meet recommended guidelines [21]. This could involve recreational activity, play, or active commuting, including light, moderate or vigorous intensity activity during a working day. The intervention had to be a defined and standardized program that could be replicable and subject to efficacy testing, excluding documents comprising solely generic encouragement to ‘sit less’ or ‘be active’.Published after 2010 in English.Conducted in a high or middle income country (HMIC) where WfH trends and occupations have concentrated [5].

Exclusion criteria:Interventions developed for clinical use, institutionalized populations, elite athletes, older adults, children, students, or non-working populations.Conference abstracts with limited information.News articles, generic advice, or factsheets.

Screening was conducted using Covidence [22]. For each document, two of the authors provided a blind rating on its potential inclusion. Any disagreements between reviewers were resolved by a third author. We then assessed the full text of documents surviving title/abstract screening for eligibility using the same criteria.

### 2.3. Data extraction and Analysis

Using an iterative approach, we extracted the following information from the included studies: country, date (pre/post COVID-19), WfH context, study population, descriptions of the intervention and outcomes, provider organization (including sector/discipline), level of scientific evidence (published or otherwise), and strength of evidence provided (i.e., study design, sample size). The information was then summarized using thematic analysis in relation to the research question.

## 3. Results

### 3.1. Overview of Data

Of 136 full texts assessed for eligibility, we excluded 123. Many of the documents, particularly from targeted websites, provided only generic information or advice, encouraging workers to do PA and be aware of SB but offering no structured actions that constituted an intervention. We excluded other documents because they focused on interventions or outcomes (e.g., work productivity) for which PA or SB may have had little direct or indirect impact (*n* = 12). Other documents did not specifically refer to WfH (*n* = 34) or relevant working populations (*n* = 3). Figure 1 describes the PRISMA flowchart.

Eleven documents were included for extraction and analysis: of these, seven were sourced from published literature databases, seven as peer-reviewed publications [23,24,25,26,27,28,29] and one document as an expanded case study of an included article [30]. Two further documents were identified using Google [31,32]. One [33] was from a targeted website. Only one of the documents identified was published pre-COVID-19 [26].

### 3.2. Intervention Characteristics

The interventions identified (Table 2) were predominantly short, structured and repetitive exercise programs which could be performed within the home working environment with minimal equipment (e.g., elastic band exercises, yoga mats) [23,25,27,31,32,33]. These interventions varied from a sequence of one-minute exercises to 50-minute video workout to be completed independently daily, three times a week or once per week (for the longer duration program). The duration of the exercise programs ran from 1–4 months. In terms of weekly PA, all six of the exercise programs ran for a duration less than 60 minutes/week of PA. One intervention was a trial of sit-stand workstations for employees while WfH [24]. This intervention included an online behavioral program of weekly education sessions that built intentions to reduce SB during the workday, and encouraged participants to set small, manageable goals to sit less and building short walks and/or activity breaks into the day. The program also prompted individuals to virtually interact with other participants. Another behavioral intervention included group and individual activities including an initial group education session, a personalized action plan, weekly email prompts and optional group support [26]. A third behavioral intervention used online educational and motivational content as well as access to a physical trainer and physiotherapist [30]. One intervention used a group social media platform to inform and motivate (goal setting, feedback and social support and social networking) PA and SB breaks during work and non-working days [29]. Activity trackers were used for self-monitoring in two interventions [24,26].

Where identified, intervention populations were office-based workers, selected from one organization in the case of four of the interventions, while three interventions recruited participants more broadly from any occupation/sector through social networks or social media recruiting methods. Participants were generally young-middle aged adult employees and female, although two studies involved primarily male workers (female ~35%).

### 3.3. Level of Scientific Evidence

Just over half of the included studies (*n* = 6) used experimental or quasi-experimental evaluation designs. Of these, five included control groups. Primary outcomes reported included total minutes of PA, three also reporting SB. The exercise programs (of a set duration of PA) recorded their impact on SB other outcomes. These outcomes included musculoskeletal discomfort, psychological outcomes, work performance and overall health. None of these studies measured total PA to determine the contribution to overall PA levels. Two studies used objective measures to record PA (accelerometers, blood biomarkers) and the other four provided only self-reported outcomes. Study populations measuring efficacy were broadly samples selected for convenience during the COVID-19 pandemic, only one collected evidence on WfH during ‘business as usual’ and while this study did not compare with a control arm it did compare workers activity while working in home and office settings [26]. Three documents, selected from unpublished sources, included no information about efficacy.

## 4. Discussion

WfH changes brought about by the COVID-19 pandemic are likely to continue in some form (either fully or partially) for many organizations and for many employees. Multiple benefits have been enabled by working from home [2]. At the same time, accumulating evidence suggests that WfH contributes to poorer levels of PA and increasing SB in home-based workers. [9,11]. Addressing these challenges requires a public health response. This does not mean that working from home should be discouraged, rather that population health actions should be implemented to enable the population WfH to do so in a way that minimizes the negative impacts on health and wellbeing, particularly on PA. As such, this review identified interventions that support PA for home-based workers and provides guidance to health authorities, employers, and advocacy groups.

### 4.1. Summary of Findings and Reflection on Existing Evidence

While the COVID-19 pandemic catalyzed many changes for working populations globally, we found surprisingly few PA interventions reported in the WfH context. Virtual exercise programs were the most common intervention type. However, none of the identified exercise programs measured total levels of PA or SB time over the working day/week, providing little evidence of the overall effects of the interventions on overall PA levels. Organized group or individually guided virtual programs are very readily available, such as the three grey literature interventions identified here, but the efficacy of these interventions for implementation by workplaces appears to be limited. Many other organic PA programs may have been promoted during the pandemic by organizations for employees in virtual group or individual settings.

The provision of sit-stand desks was effective, both in interrupting SB and in improving PA in one study but did not result in changes in cardiometabolic health [24]. The existing evidence here is limited but suggestive that active workstations allow for more physical activity [34]. The demand for active workstations may have increased substantially from the start of the pandemic but the effects on physical health outcomes in any context have been small [35]. More home-based studies on office infrastructure including sit-stand desks are needed given the heterogeneity of home environments. These types of simple interventions are low resource intensive and may have greater potential for rapid adaptation to home contexts.

Four interventions identified were behavioral type interventions offering health education, behavioral support, motivational counselling, goal setting, and peer support through virtual platforms. Activity trackers, emails or other prompts were also used to encourage PA or break SB. In general, multi-component workplace interventions have been the most effective in improving health behaviors in workplace settings [18].

Only one intervention explored workers behavior both in the home and office environment and it appeared to be more effective in changing SB when workers attended the office than at home [26], suggesting barriers in the home environment. The duration of the interventions identified were limited to a few months (four at the most), therefore it is not known how long any of the interventions suggested would sustain PA implemented over time in WfH populations.

### 4.2. Implications for Research

Evidence on effective interventions to support PA in the physical workplace setting is well established [15,16,17,18], but evidence gaps have emerged on how existing effective programs or new interventions may contribute to PA in the WfH context. Efficacy and effectiveness trials need to include more objective markers of PA to determine the impact of such interventions on total PA levels.

Some recommendations during the initial phase of the pandemic promoted active work meetings, standing desks for home-based work, and incentives to encourage employees to exercise [36]. These interventions need to be evaluated to determine the contribution of interventions to increases in PA and reductions in SB, and to build further evidence in demonstrating the contribution to meeting recommended PA levels for health. Most of the studies in this review were conducted during the initial phases of the COVID-19 pandemic when WfH was more likely to be enforced. This means that new long-term PA interventions need to be developed and trialed for the current post-pandemic context, including those adapted for hybrid models of working. This may involve face-to-face and online program components and more varied programs to allow for multiple settings and levels of interaction. To improve the evidence base, interventions co-designed by researchers, employees and managers may ensure that interventions are both effective and sustainable [37].

Implementation research evidence is needed to improve understanding about the contextual factors influencing the successful uptake of these initiatives in the home environment. Implementation challenges in office settings include the workplace culture, job requirements and public perceptions [37]. These may remain as a challenge even in WfH context. The blurring of work-home boundaries in WfH contexts may create other challenges that require qualitative and quantitative research investigation. For example, structured programs, group support and guidance may be more suitable for certain individuals and their home context, and unstructured PA support interventions more suitable for other employees. The barriers and enablers to implementing and sustaining these WfH interventions, as well as their effectiveness in sub-populations needs to be explored. In addition, a broader research agenda, moving beyond workplace boundaries, is needed to address the new challenges of a rising and more permanent WfH population.

### 4.3. Implications for Policy and Practice

WfH will continue to grow and evolve into new labor markets and sectors including in lower income countries in the future. Supporting equity of access will require a suite of flexible interventions which can accommodate home-based, office-based and hybrid working. WHO recommends employers implement programs to promote healthy WfH, stating the importance of routine breaks and the need for achieving sufficient PA, but stops short on specifying how that could be achieved [10]. The OECD provides guidance on policies for improving WfH productivity, but without any connection to health [5]. Given that national guidance specifically for WfH is also limited, a gap in leadership and knowledge on how to promote PA in WfH is evident [6,38,39]. Recommendations of which types of interventions organizations need to implement is needed. At the least, these interventions should be multicomponent, able to be implemented in multiple settings and encourage PA during the work day and breaks from SB. The implications of not providing more structured guidance for workplaces may have a ripple effect. The most recent global status report on PA indicates that one-third of countries around the world have implemented national policies promoting PA in workplaces [40]. If observed patterns of poorer PA in the working population persist, then it will be more difficult to achieve the WHO’s 2030 population targets for PA [14].

Finding ways to support PA within and beyond the working day is central to preventing chronic diseases in the adult population. Incidental PA accrued throughout the day by workers walking to a meeting or to food outlets as well as too and from work may be lost if home-based workers do not leave their homes (or do so by car). Alternative forms of PA are needed to replace these losses. Commuting to work by active or public transport five days per week can provide adults with sufficient PA for health [41]. Active travel is not restrained to the work commute and can be undertaken for other purposes. However, while promoted in many countries active travel remains limited. In the US and Australia, for example, passive commuting is high [42,43,44] and increasing in other countries, like China [45]. WfH may provide time saving for PA that may have been lost by a passive commute, providing benefits to this proportion of the population however only if the time is allocated to PA is not switched to other passive activities. Great variations in PA occurred across sub-populations WfH under enforced COVID-19 restrictions, suggesting that some groups adapt but others will need greater support [46]. Without concerted, inclusive efforts to support PA in the adult working population, physical activity and other health disparities may increase [14].

Changes to workplace policies and enhanced urban planning is needed, with policies fostering a culture and norms that prioritize health and wellbeing of employees. A combination of individual, organizational and environmental level strategies may be effective, particularly for reducing workplace SB [18]. Organizational and/or management commitment to support PA during work hours is essential. In the WfH context, strategies that reconnect isolated workers with each other may be important to support and maintain behavior change. Short-term incentives, such as health fund rewards or gym memberships could initiate long-term behavior change [47], however interventions outside the workplace have had little previous impact on sedentary behavior [48].

The pandemic has highlighted the issue of equitable access to environments supportive of PA, including greenspace and walking/cycling infrastructure for recreational and active travel [49]. The built environment facilitates PA [50,51]. Planning and designing cities and neighborhoods around the ‘15-minute sustainable city’ model, such that many day-to-day activities, including work, can be accessed locally by walking and cycling will provide many benefits for people WfH achieving PA. In many cities cycling, walking and use of greenspace increased during the pandemic [52,53], suggesting this aspiration is realistic.

## 5. Limitations

While this review has sought to comprehensively investigate PA interventions for WfH, we excluded interventions which did not explicitly refer to the WfH environment. Similarly, as we were interested in the working adult population, we excluded studies conducted to support PA in school and university populations. The paucity of research studies we found that focused on WfH adult contexts is a major evidence gap and limits any conclusive assessment of the types of interventions that should be implemented.

## 6. Conclusions

Work culture has changed as a legacy of the COVID-19 pandemic with more people WfH. Few interventions have been developed and evaluated to support PA and interrupt SB for the WfH context, suggesting a need to adapt and test more interventions for PA promotion/ SB interruption in the WfH setting. The existing evidence is inadequate to determine the potential impact of PA interventions for WfH settings. Therefore, to reduce the risk of chronic diseases associated with physical inactivity there is a need to improve knowledge and practice-based interventions, and this should remain a focus of population health into the future as business models for WfH emerge and evolve.

## Figures and Tables

**Figure 1 ijerph-20-00073-f001:**
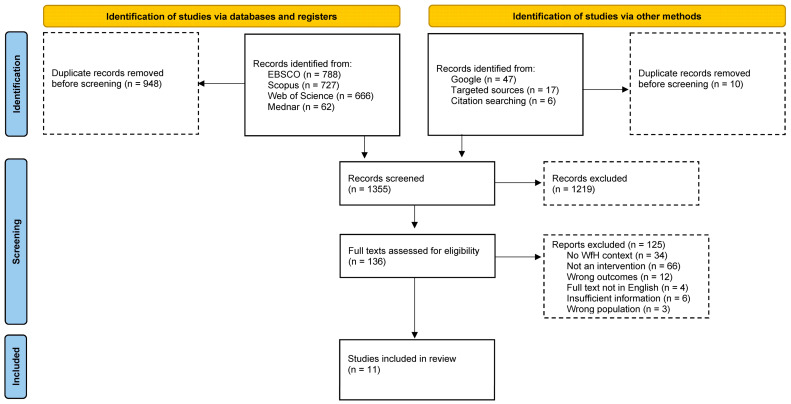
Flowchart of published and grey literature investigated.

**Table 1 ijerph-20-00073-t001:** Summary search strategy.

Published Literature Databases	Scopus, Web of Science, Ebsco Host (Rehabilitation & Sports Medicine Source, MEDLINE Complete, Psychology and Behavioral Sciences Collection, Applied Science & Technology Source Ultimate, Business Source Ultimate, CINAHL Complete, SPORTDiscus)
Search Engines	Google; Mednar
Targeted Websites	*Business consulting* (Deloitte; Gartner; Forbes; Earnest & Young; Price Waterhouse Cooper; McKinsey & company) *Government authority* (Australian Productivity Commission; SafeWork Australia; National Institute for Health & Care Excellence UK, Health & Safety Executive UK; Centre for disease control USA) *International authority* (World economic forum; World health organisation)
Search Terms	(telecommut* OR “work* from home” OR “work* at home” OR telework* OR ( workplace* N1 flexib* ) OR “flexible work* arrangement*” OR “home working” OR “homeworking” OR “virtual office” OR “stay* at home”) AND ( physical* activ* OR “active travel*” OR exercise OR “sedentary behavior” OR sitting OR fitness OR sport* OR walk* OR cycling OR bicycl* OR “energy expenditure” OR “aerobic train*” OR “physical exertion” OR “resistance train*” OR “strength train*” ) AND AB ( intervention* OR initiative* OR experiment* OR evaluat* OR program* OR promot* OR policy OR strategy OR “prospective stud*” ) AND health*

**Table 2 ijerph-20-00073-t002:** Summary of included documents.

First Author and Reference	Country	Study Design	Intervention Description	Population	Sample Size	Primary Outcome(s) and Measures
Garcia [23]	Ecuador	Pseudo-RCT ^	A one-month yoga program of daily 10-min. routines.	Desk-based workers (>18 years) (65% female)	*n* = 94 (54 intervention; 40 control)	Self-reported musculoskeletal discomfort, mood disturbance
Mailey [24]	United States	RCT	Provision of sit-stand desks; online education program designed to set goals and motivate SB breaks and PA.	University workers (>18 years) from one university (80% female)	*n* = 95 (24 intervention desk only; 23 programs only; 23 desk + program; 25 control)	Self-reported sitting time, total PA minutes (recorded by accelerometer) (Mets), BMI, cardiometabolic biomarkers (blood samples)
Muniswamy [29]	India	RCT	Social media-based intervention 8 weeks—Facebook group with tailored information on PA (scheduled exercises, demonstrations, advice) and mental health (relaxation, work breaks), peer group motivation chat page, individual access to expert advice	Healthy young IT professionals (23–35 years) with active social media use; poor mental/physical health (35% female).	*n* = 46 (22 intervention; 26 control)	MVPA, sitting time, frequency of SB breaks, aerobic activity (VO_2_), BMI, stress, anxiety and depression scores
Moreira [25]	Portugal	Quasi-experiment	Workplace exercises with 3 × 15 min. sessions/wk. for 4 months.	Office workers from an automotive organisation (18–65 years; 36% female)	*n* = 39 (13 intervention; 26 control)	Self-reported health status (SF-36 measuring pain, physical functioning, physical performance, emotional performance)
Núñez-Sánchez [28,30]	Spain	Qualitative + post-intervention observation	Two corporate policies and behavioral change intervention with a PA component (online trainer, tutorial videos, motivational videos, classes for family sports, online physiotherapist, and fitness and physio content).	Office workers from one multi-location organisation	*n* = 255 (253 survey participants; 2 interviewees)	Self-reported change in PA; program satisfaction
Olsen [26]	Australia	Quasi-experiment (no control group)	Behavioral change (group session for goal setting, and action planning); personalized activity plan, provision of an activity tracker for self- monitoring; social support; weekly email prompts; educational seminar.	Office workers (mean age 39.5 years) from one organisation (69% female)	*n* = 30 (within case comparison of office and WFH days)	Self-reported sitting time; total sitting time/day; Total PA minutes/day (walking & MVPA) (recorded by accelerometer)
Wadhen [27]	United Kingdom	RCT	50-min online yoga classes every week for 6 weeks.	Workers from unspecified occupations (mean age 42.5 years) years) London-based organisations (female 91%)	*n* = 52 (26 intervention; 26 control)	Self-reported stress, depression, anxiety & self-efficacy; perceptions
Vantage Fit [32]	Global	None	Online exercise program of 7 exercises for daily duration >15 min.	Not stated	n/a	Not stated
NIVA Education [31]	Denmark	None	Online exercise program selection of 3 videos of 10 min. duration, recommended 3 × week.	Not stated	n/a	Not stated
Be Upstanding [33]	Australia	None	Online exercise program set of 15 exercises with reps., designed to interrupt SB (suggested every 30 min). No duration specified.	Not stated	n/a	Not stated

^ Some participants self-selected. BMI = Body mass index. IT—information Technology. MVPA = moderate–vigorous PA. RCT = randomized control trial.

## Data Availability

No new data was created or analysed in this study. Data sharing is not applicable to this article.

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
