# Peer review of "Interventions Designed to Support Physical Activity and Disease Prevention for Working from Home: A Scoping Review"

_ijerph, 2022, doi:10.3390/ijerph20010073_

Round 1

Reviewer 1 Report

see file attached

Reviewer 2 Report

This paper presents a scoping review of physical activity interventions for individuals working from home. the paper is generally well-written and is a very timely topic, given the COVID-19 pandemic and changing working conditions around the globe. That said, I finished the review feeling like I'd missed some pages. It was extremely brief, and, aside from the one table, there was no description of any of the studies or their findings. The discussion section is well written, but discussed the field in general and where it should go rather than discussing the findings from this actual review. Granted, this is a scoping review not a systematic review, but more exploration of the included studies and discussion of their findings would be helpful.     Introduction: -the introduction is very well written, and highlights what a timely topic this is   Methods: -could the authors explain why they limited the studies to those in high-middle income countries?   Results: -the authors have provided a summary of the studies, but no information about the common outcomes or measures used; there is very little information presented about the actual studies. Since there are so few, the authors could provide a more detailed overview of them. As is, there is relatively no information provided about the studies themselves or their findings   Discussion: -the discussion is well written, but focuses mostly on the research context and future directions. As is, there is very little discussion of the actual findings from the review 

Reviewer 3 Report

This paper is fine and worth publishing as it is.

Author Response

Thank you. We have expanded the results and discussion sections of the paper as suggested by Reviewers 1 and 2 and have attached it for your persual. 

Round 2

Reviewer 2 Report

The authors have done a commendable job revising the paper. They have adequately addressed my comments.